# Interaction between Two Iridovirus Core Proteins and Their Effects on Ranavirus (RGV) Replication in Cells from Different Species

**DOI:** 10.3390/v11050416

**Published:** 2019-05-04

**Authors:** Xiao-Tao Zeng, Qi-Ya Zhang

**Affiliations:** 1State Key Laboratory of Freshwater Ecology and Biotechnology, Institute of Hydrobiology, Wuhan 430072, China; zengxt@ihb.ac.cn; 2The Innovation Academy of Seed Design, Chinese Academy of Sciences, Beijing 100101, China

**Keywords:** *Rana grylio* virus (RGV), iridovirus core proteins, protein interaction, aquatic animals, cross-species transmission, yeast two-hybrid (Y2H), co-immunoprecipitation (Co-IP)

## Abstract

The two putative proteins RGV-63R and RGV-91R encoded by *Rana grylio* virus (RGV) are DNA polymerase and proliferating cell nuclear antigen (PCNA) respectively, and are core proteins of iridoviruses. Here, the interaction between RGV-63R and RGV-91R was detected by a yeast two-hybrid (Y2H) assay and further confirmed by co-immunoprecipitation (co-IP) assays. Subsequently, RGV-63R or RGV-91R were expressed alone or co-expressed in two kinds of aquatic animal cells including amphibian Chinese giant salamander thymus cells (GSTCs) and fish *Epithelioma papulosum cyprinid* cells (EPCs) to investigate their localizations and effects on RGV genome replication. The results showed that their localizations in the two kinds of cells are consistent. RGV-63R localized in the cytoplasm, while RGV-91R localized in the nucleus. However, when co-expressed, RGV-63R localized in both the cytoplasm and the nucleus, and colocalized with RGV-91R in the nucleus. 91R△NLS represents the RGV-91R deleting nuclear localization signal, which is localized in the cytoplasm and colocalized with RGV-63R in the cytoplasm. qPCR analysis revealed that sole expression and co-expression of the two proteins in the cells of two species significantly promoted RGV genome replication, while varying degrees of viral genome replication levels may be linked to the cell types. This study provides novel molecular evidence for ranavirus cross-species infection and replication.

## 1. Introduction

A large number of aquatic viruses regulate population dynamics and community interactions in aquatic ecosystems [1]. They are also involved in aquatic animal diseases. As pathogens, aquatic animal viruses often infect shellfishes, fishes, amphibians, reptiles and aquatic mammals [2,3,4,5]. Ranaviruses (*Rana grylio* virus, RGV) are members of the family *Iridoviridae* that are large, double-stranded DNA viruses and infect ectothermic vertebrates [6]. Importantly, ranaviruses are capable of crossing species barriers of numerous ectothermic vertebrates and can spread between different species [7,8,9]. Recent reports have revealed more about the molecular mechanism of aquatic viral disease, including ranaviral disease [10]. Science and technology have been applied to a wide range of studies on aquatic viruses [11]. Different approaches, such as co-immunoprecipitation (co-IP) assays [12], fluorescence microscopy, and yeast two hybrid (Y2H), have been widely used in the investigation of protein–protein relationships or interactions [13]. Recently, the Y2H assay was used to analyze protein–protein interactions among the structural proteins of *Chilo* iridescent virus, a member of genus *Iridovirus* [14].

DNA polymerases play multiple roles [15]. Its main function is DNA replication and is capable of catalyzing DNA synthesis [16]. DNA polymerases of some large DNA viruses, such as herpesvirus and vaccinia virus, play a crucial role in virus genome replication [17,18]. Iridoviral DNA polymerases (RGV-63R and its homologous proteins) are believed to be essential components for virus DNA replication [19,20]. RGV-91R and its homologous proteins are considered proliferating cell nuclear antigens (PCNAs). This kind of protein is also found in humans as a cofactor of DNA polymerase, which can increase the processivity of DNA strand synthesis during replication [21]. In eukaryotes, PCNA as a sliding clamp protein forms a ring-shaped homo-trimer that encircles double-stranded DNA. It can confer high processivity with respect to replicative DNA polymerase. Moreover, it forms a mobile platform on DNA that recruits many of the proteins involved in DNA replication, repair, and recombination. For example, PCNA interacts with DNA polymerase δ, a member of family B [22,23]. There are 26 core proteins shared in *Iridoviridae* [24], including RGV-63R, RGV-91R, and some other proteins related to transcription, replication, and nucleotide metabolism. Several RGV encoded proteins involved in gene transcription, viral infection, and assembly have been identified previously [25,26,27,28].

However, the ranavirus protein–protein interactions and their effects on virus replication in cross-species transmission remain unaddressed. The goal of this study was to investigate the interactions between ranavirus core proteins, and their effects on virus replication in both fish cells (*Epithelioma papulosum cyprinid* cells, EPCs) and amphibian cells (Chinese giant salamander thymus cells, GSTCs) through multiple technical approaches, such as Y2H, co-IP assays, fluorescence microscopy, Western blotting and qPCR. Y2H was used to test that the gene products can physically bind to each other, while a fluorescence experiment was used to test whether the gene products had similar co-localization in different host cells. Co-IP was performed to confirm the results from Y2H.

## 2. Materials and Methods

### 2.1. Cell Lines and Virus

*Epithelioma papulosum cyprinid* cells (EPCs) and Chinese giant salamander thymus cells (GSTCs) [29] were cultured in Medium 199 supplemented with 10% fetal bovine serum (FBS) at 25 °C. They were used for all work needing cells, except the co-immunoprecipitation (co-IP) assay. *Rana grylio* virus (RGV) was used in the study and the virus was repeatedly cultured on EPCs for long-term preservation. Human embryonic kidney (HEK293T) cells were grown at 37 °C in 5% CO2 in Dulbecco’s modified Eagle’s medium (DMEM, gibco, Thermo Fisher Scientific, Waltham, MA, USA) supplemented with 10% FBS, and were only used for the co-immunoprecipitation (co-IP) assay.

### 2.2. Plasmid Construction

RGV propagation in EPC cell cultures, purification, and its genomic DNA preparation were performed as described previously [27]. Different kinds of nucleic-acid fragments were amplified with the corresponding primers (Table 1). The plasmids used for yeast two-hybrid (Y2H), co-immunoprecipitation (co-IP), fluorescence microscopy and quantitative polymerase chain reaction (qPCR) were constructed. All constructs used in this study were confirmed by DNA sequencing. The operations are as follows:

To construct the bait plasmid for screening the interacting protein of RGV-63R, the gene that encodes RGV-63R was amplified with primers *63R*-F/R and cloned into pGBKT7 to produce bait plasmid pGBKT7-*63R*. To construct the prey plasmids containing 26 RGV core genes (x represents the genes, such as *91R*, *97R*, *98R,* and *102R*), fragments were amplified from RGV genomic DNA and cloned into pGADT7 to obtain prey plasmids pGADT7-x.

To generate the plasmids for coimmunoprecipitation (co-IP) assays, *RGV-63R* was amplified with primers *63R*-F2/*63R*-*3Flag*-R and cloned into pcDNA3.1(+) to construct pcDNA3.1-*63R*-*3Flag*. *RGV*-*91R* was amplified with primers *91R*-F2/*91R-HA*-R and cloned into pcDNA3.1(+) to construct pcDNA3.1-*91R-HA*.

To analyze the colocalization between RGV-63R and RGV-91R, the *RGV-63R* was amplified with primers *63R*-F/*63R*-R3 and cloned into pEGFP-N3 to construct pEGFP-*63R*. *RGV-91R* was amplified with primers *91R*-F3/*91R*-R2 and cloned into pDsRed2-C1 to produce pDsRed2-*91R*.

The RGV-91R (245aa) sequence and structure domains are as follows. Grey shades indicate nuclear localization signals (NLS1 and NLS2):

MLWEAVTDKPVKLKGLLELLLNNMDSARLVVTSQSVSVVDYQSNMAVTASMPSSVFTSYVYKSDAECLYAGLPHAALPDLKSFKAKCNVTLRLMGDPECGQYTMKIIIANASHMSTSINMVVDHGKKEADRGHPEGAGKSFTLTQQEFNTLCKTFKQGPVNLGVFGGVLVASGGVDGIKVKEVAFGAPDCVTPHVKLCVHAEKMSRLVKMGPFSAGSLTVCVAQGSVTVSTRGHLGSLTVTLFEG.

To analyze the effect of the nuclear localization signal (NLS) on the subcellular location of RGV-91R, two DNA fragments *91R-N1* and *91R-C1* were amplified from RGV genomic DNA with primers *91R-F3*/*91R△NLS1-R* and *91R△NLS1-*F/*91R-*R2, respectively. The DNA fragment *91R△NLS1* was amplified via overlap PCR using *91R-N1* and *91R-C1* as templates and cloned into plasmid pDsRed2-C1 to produce pDsRed2-*91R△NLS1*. As described above, primers *91R-*F3/*91R△NLS2-*R and *91R△NLS2-*F/*91R-*R2 were used to construct the plasmid pDsRed2-*91R△NLS2*.

The previously constructed plasmid pMD-*MCP* [30] was used for a standard curve of qPCR.

### 2.3. Yeast Two Hybrid (Y2H) Assays

The RGV-63R gene was cloned into bait vector pGBKT7 and expressed in the yeast Y2HGold strain and cultured on SD/-Trp plates. Twenty-six RGV core genes (x represents) including *91R*, *97R*, *98R,* and *102R*, which were considered as iridovirus core genes, were cloned into prey vectors pGADT7 and expressed in the yeast Y187 strain, and cultured on SD/-Leu plates. However, no colony formed when yeast Y187 was separately transformed with pGADT7-*73L*/*95R*/*101L*. The interactions between bait protein pGBKT7-63R and 23 prey proteins pGADT7-x were then identified using the yeast two-hybrid (Y2H) system and were performed according to the Matchmaker Gold Yeast Two-Hybrid System User Manual (Clontech, Clontech Laboratories, Inc., Mountain View, CA, USA). Three parallel experiments were performed. Y2HGold/pGBKT7-*63R* and Y187/pGADT7-x were placed in 2×YPDA media and incubated at 30 °C for 24 h. The mating yeast cells were collected by centrifugation, and resuspended with 0.5×YPDA media, and they were then cultured on 2Q (SD/-Trp/-Leu) and 3Q/X (SD/-Trp/-Leu/-His/X-α-Gal) plates for 6 days at 30 °C. As a positive control, yeast mating between Y187/pGADT7-*T* and Y2HGold/pGBKT7-*P53* was carried out in the same way, since T interacts with P53. As a negative control, yeast mating between Y187/pGADT7-*T* and Y2HGold/pGBKT7-*Lam* was carried out in the same way, since T does not interact with Lam. Interactions take place if colonies form on both 2Q and 3Q/X plates. As a positive control, many colonies formed on 2Q and 3Q/X plates, and as a negative control, no colonies formed on 3Q/X.

### 2.4. Co-IP Assays

To further confirm interaction between RGV-63R and RGV-91R detected by Y2H, co-IP assays were performed. Briefly, HEK293T cells seeded in 10 cm dishes were cotransfected with 5 μg of pcDNA3.1-*91R-HA* and 5 μg of pcDNA3.1-*63R-3Flag*. As a control, 5 μg of pcDNA3.1-*91R-HA* and 5 μg of empty vector pcDNA3.1 were cotransfected in parallel. At 24 h post-transfection (hpt), cells were lysed with a radio immunoprecipitation assay (RIPA) buffer containing protease inhibitor cocktail (Roche), and cell lysates were incubated with Red Anti-HA Affinity Gel (Sigma, Sigma-Aldrich LLC., Santa Clara, CA, USA) overnight at 4 °C. The precipitates were collected by centrifugation, washed with ice-cold phosphate-buffered saline (PBS) three times, eluted with 40 μL of PBS, and finally subjected to Western blot analysis.

### 2.5. Western Blot Analysis

Western blot analysis was performed as described previously [30]. Protein samples were resolved by SDS-PAGE, followed by electroblotting to polyvinylidene difluoride (PVDF) membrane. The blots were probed with anti-HA mouse monoclonal antibody (1: 1000, Sigma, Sigma-Aldrich LLC., Santa Clara, CA, USA) or anti-Flag rabbit monoclonal antibody (1: 1000, Abbkine, Abbkine Scientific Co., Ltd. Wuhan, China). Peroxidase-conjugated goat anti-mouse or anti-rabbit IgG (H+L) antibody was used as the secondary antibody. The signals were detected with a chemiluminescent horseradish peroxidase (HRP) substrate (Millipore, Millipore Corporation, Billerica, MA, USA).

### 2.6. Fluorescence Microscopy

Colocalization assays were performed to confirm the interaction between RGV-63R and RGV-91R. EPCs or GSTCs transfected with plasmid pEGFP-*63R* or pDsRed2-*91R* alone or cotransfected with both plasmids were fixed, permeabilized, stained with Hoechst 33342, and observed under a Leica DM IRB fluorescence microscope (objective 100×), as described previously [31].

### 2.7. Quantitative Analysis of RGV Genomic DNA

To investigate the effects of RGV-63R and RGV-91R on RGV replication in cells from different species, the DNA level of RGV in EPCs or GSTCs expressing one protein alone or coexpressing both proteins were measured by qPCR, respectively. A total of 2 × 10^5^ EPCs or GSTCs grown in 24-well plates were transfected with 0.5 μg of empty vector pcDNA3.1 or cotransfected with 0.25 μg of pcDNA3.1-*63R*-*3Flag* and 0.25 μg of pcDNA3.1, 0.25 μg of pcDNA3.1-*91R*-*HA* and 0.25 μg of pcDNA3.1, or 0.25 μg of pcDNA3.1-*63R*-*3Flag* and 0.25 μg of pcDNA3.1-*91R*-*HA*. The transfected cells at 24 hpt were harvested and subjected to Western blot analysis. For qPCR, at 24 hpt, the transfected cells were infected with RGV with an MOI of 0.5. The infected cells at 48 h post-infection (hpi) were harvested and washed with PBS three times. The viral genomic DNA was then extracted according to the instructions of TaKaRa MiniBEST Universal Genomic DNA Extraction Kit. Each DNA sample was dissolved in 100 μL of elution buffer, and 1 μL of each DNA sample was used for each qPCR reaction. Specific primers *MCP-*F/*MCP-*R for RGV MCP gene was used for qPCR. qPCR was carried out using Fast SYBR Green Master Mix with the StepOneTM Real-Time PCR System (Applied Biosystems, Foster City, CA, USA), as described previously [32].

## 3. Results

### 3.1. Interaction between RGV-63R and RGV-91R was First Screened by Y2H

The interactions between RGV-63R and other 23 proteins from RGV were analyzed by yeast two hybrid. The results of detecting interactions between RGV-63R and 91R, 97R, 98R, or 102R are shown in Figure 1. Only RGV-91R showed interaction with RGV-63R. When the interaction between RGV-63R and RGV-91R was detected, colonies formed on SD/-Trp/-Leu (2Q) and SD/-Trp/-Leu/-His/X-α-Gal (3Q/X) plates. When detecting the interactions between RGV-63R and 97R, 98R, or 102R, colonies formed on 2Q plates, but no colony formed on 3Q/X plates.

### 3.2. Confirmation of Interaction between RGV-63R and RGV-91R by Co-IP Followed by Western Blotting

Co-IP assays were used to confirm the interaction between RGV-63R and RGV-91R. The proteins 63R-3Flag and 91R-HA were co-expressed and anti-HA antibody affinity gel was used to precipitate the protein complexes. The result of co-IP followed by western blotting is shown in Figure 2; 63R-3Flag and 91R-HA were detected in cell lysates and immunoprecipitated (IP) protein complexes from HEK293T cells cotransfected with pcDNA3.1-*63R*-*3Flag* and pcDNA3.1-*91R*-*HA*. However, only 91R-HA was detected in cell lysates and immunoprecipitated protein complexes from pcDNA3.1 and pcDNA3.1-*91R*-*HA* cotransfected cells. This further confirmed that RGV-63R interacts with RGV-91R.

### 3.3. Localization of RGV-63R and RGV-91R in Cells of Different Species

Fluorescence microscopy was carried out. The protein 63R-EGFP expressed alone localized in the cytoplasm of GSTCs or EPCs. The protein 91R-RFP expressed alone localized in the nucleus of GSTCs or EPCs. Obviously, the localizations of the same protein in GSTCs or EPCs from different species were consistent (Figure 3A).

When 63R-EGFP and 91R-RFP were co-expressed, they colocalized in the nucleus of GSTCs. They consistently expressed in EPCs, as shown in Figure 3B. These results not only further confirmed the interaction between RGV-63R and RGV-91R, which alters RGV-63R localization, but also exactly showed the same phenomenon that was observed in cells from two different species. These results indicate that the two proteins of RGV can play the same role in the cells of different species.

RGV-91R was predicted to contain two NLSs, namely, NLS1 (60~88 aa) and NLS2 (176~207 aa). When the protein 63R-EGFP was co-expressed with NLS deletion proteins (91R△NLS1-RFP or 91R△NLS2-RFP), they colocalized in the cytoplasm rather than in the nucleus (Figure 3B). These revealed that the NLS domain of RGV-91R is essential to transporting RGV-63R into the nucleus.

### 3.4. Detection of RGV-63R and RGV-91R Expressions by Western Blotting

The plasmids pcDNA3.1-*63R*-*3Flag* and pcDNA3.1-*91R*-*HA* were mixed when they were cotransfected into cells. The transfection efficiency was about 10%. There is about 7.6% cells cotransfected with plasmids pcDNA3.1-*63R*-*3Flag* and pcDNA3.1-*91R*-*HA*, and about 2.4% cells transfected with alone plasmid by counting the number of transfected cells using immunofluorescence.

The expressions of RGV-63R and RGV-91R in GSTCs or EPCs at 24 hpt were detected by Western blot analysis. The results were shown in Figure 4. RGV-63R and RGV-91R were detected in GSTCs or EPCs expressing one protein alone or co-expressing two proteins, indicating that RGV-63R and RGV-91R were both successfully expressed.

### 3.5. RGV-63R and RGV-91R Promote RGV Genome Replication

The effect of RGV-63R or RGV-91R on RGV genome replication was analyzed by expressing one protein alone or co-expressing two proteins in GSTCs and EPCs, respectively. qPCR analysis showed that RGV genomic copies in both GSTCs and EPCs expressing RGV-63R or RGV-91R alone or co-expressing the two proteins in EPCs were significantly higher than those in the control (Figure 5). A similar promoting activity is indicated when the two proteins are expressed alone in cells from different species.

The RGV genomic copies in EPCs co-expressing the two proteins were higher than those expressed alone. However, the genomic copies were lower when the two proteins were co-expressed rather than expressed alone in GSTCs (Figure 5), which was in line with expectations because of the long-term preservation of RGV in EPCs, not in GSTCs. This suggests that ranaviruses cross-species transmission were closely related with their adaptability to the host, but how it is affected is still an important unresolved area that needs to be studied.

## 4. Discussion

In this study, the interactions of RGV-63R with other iridoviral core proteins from RGV were tested by a Y2H assay, showing that RGV-63R interacts with RGV-91R. However, only white colonies formed on both SD/-Trp/-Leu and SD/-Trp/-Leu/-His/X-α-Gal plates while detecting the interaction between RGV-63R and RGV-91R by Y2H. It is possible that the interaction between RGV-63R and RGV-91R in yeast is weak and only activates the reporter gene *HIS3*, but does not activate the reporter gene *MEL-1* which encodes α-galactosidase hydrolyzing chromogenic substrate X-α-Gal. Subsequently, the interaction between the two proteins was further confirmed by co-IP.

In this study, 63R-EGFP protein only localized in cytoplasm of GSTCs or EPCs transfected with pEGFP-*63R.* When 63R-EGFP and 91R-RFP were co-expressed in GSTCs or EPCs, 63R-EGFP localized in both cytoplasm and the nucleus and colocalized with 91R-RFP in the nucleus. This not only further confirmed the interaction between the two proteins, but also implied that the nuclear import of RGV-63R mediated by RGV-91R might be very important for the virus genome synthesis within the nucleus. Nucleus import of macromolecules larger than 60 kDa is an active, energy-dependent process mediated by sequence-specific motif and NLSs [33]. RGV-63R was predicted to encode DNA polymerase with a predicted molecular mass of 114 kDa. This implied that the nucleus import of RGV-63R might be mediated by NLSs. Fluorescent microscopy showed that the 91R△NLS-RFP protein localized in cytoplasm and did not mediate the nuclear import of RGV-63R, suggesting that the nuclear import of RGV-63R was dependent on the NLSs of RGV-91R.

Iridoviral DNA polymerase has been considered to be involved in virus genome replication [19]. Iridoviral PCNA was also considered to be involved in nucleic acid synthesis [34]. RGV-91R was identified as a late protein, and its homologue in a member of genus *Ranavirus* was identified as a structural protein [35,36]. In this study, the effects of RGV-63R, RGV-91R, and the interaction between the two proteins on RGV replication in GSTCs or EPCs were further analyzed. The results showed that overexpression of RGV-63R or RGV-91R in both GSTCs and EPCs significantly increased RGV genome copies. This indicated that RGV-63R and RGV-91R both promote RGV replication.

An interesting phenomenon is that the number of RGV genomic copies was lower in GSTCs when the two proteins were coexpressed compared to proteins expressed alone, but it was higher in EPCs when the two proteins were coexpressed. It has been reported that some host proteins inhibit ranavirus replication [37]. It has also been reported that antiviral activities of some cellular proteins are species-specific [38,39]. Whether co-expression of the two proteins in GSTCs or EPCs is associated with these similar host proteins needs further research. It also may result from the preparation of virus stocks. Compared with virus replication in EPCs, the virus was always multiplied in GSTCs, implying that the RGV genome replication in GSTCs might be more efficient. However, more experiments are required to confirm and explain this mechanism.

In conclusion, this report is the first confirmation and description of ranaviral protein–protein interactions that promote virus genome replication in cells from different species, which provides novel molecular evidence and insights for ranavirus cross-species infection.

## Figures and Tables

**Figure 1 viruses-11-00416-f001:**
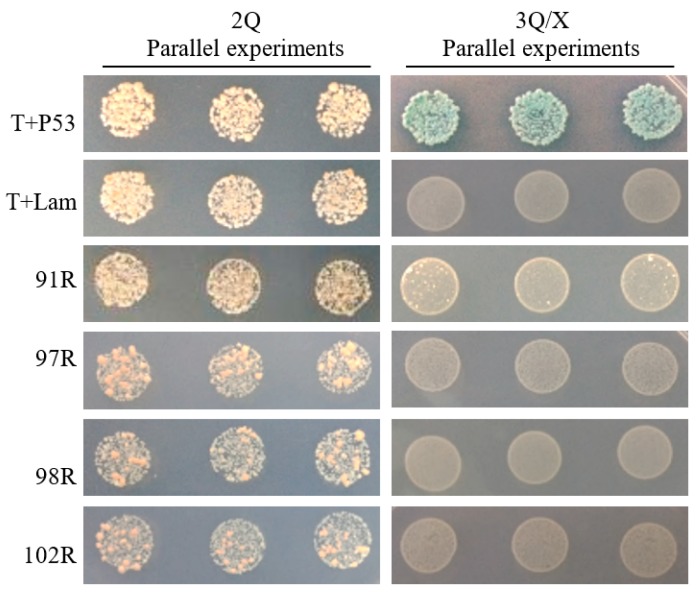
The formed yeast colonies during testing interactions between RGV-63R and four proteins by Y2H. Three parallel experiments were performed. 2Q: SD/-Trp/-Leu. 3Q/X: SD/-Trp/-Leu/-His/X-α-Gal. 91R, 97R, 98R, and 102R: RGV proteins, which are encoded by iridovirus core genes, respectively. T+P53: the positive control; T+Lam: the negative control.

**Figure 2 viruses-11-00416-f002:**
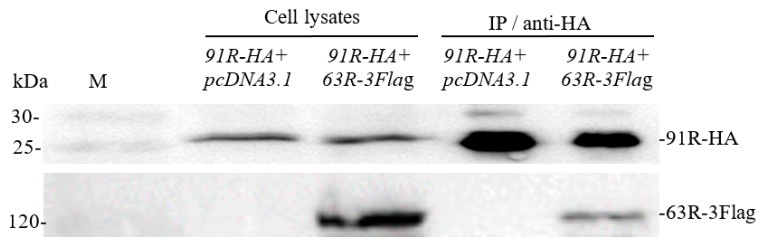
Western blot analysis for sample of testing interaction between RGV-63R and RGV-91R by co-IP. Cell lysates from HEK293T cells cotransfected with indicated plasmids (*91R-HA+ pcDNA3.1*, *91R-HA+ 63R*-*3Fla*g) and IP (immunoprecipitated) protein complexes with indicated plasmids (*91R-HA+ pcDNA3.1*, *91R-HA+ 63R*-*3Fla*g) are subjected to Western blot analysis using anti-HA and anti-Flag. Cells lysates and IP showed the bands of 91R-HA and 63R-3Flag. M: protein molecular mass marker.

**Figure 3 viruses-11-00416-f003:**
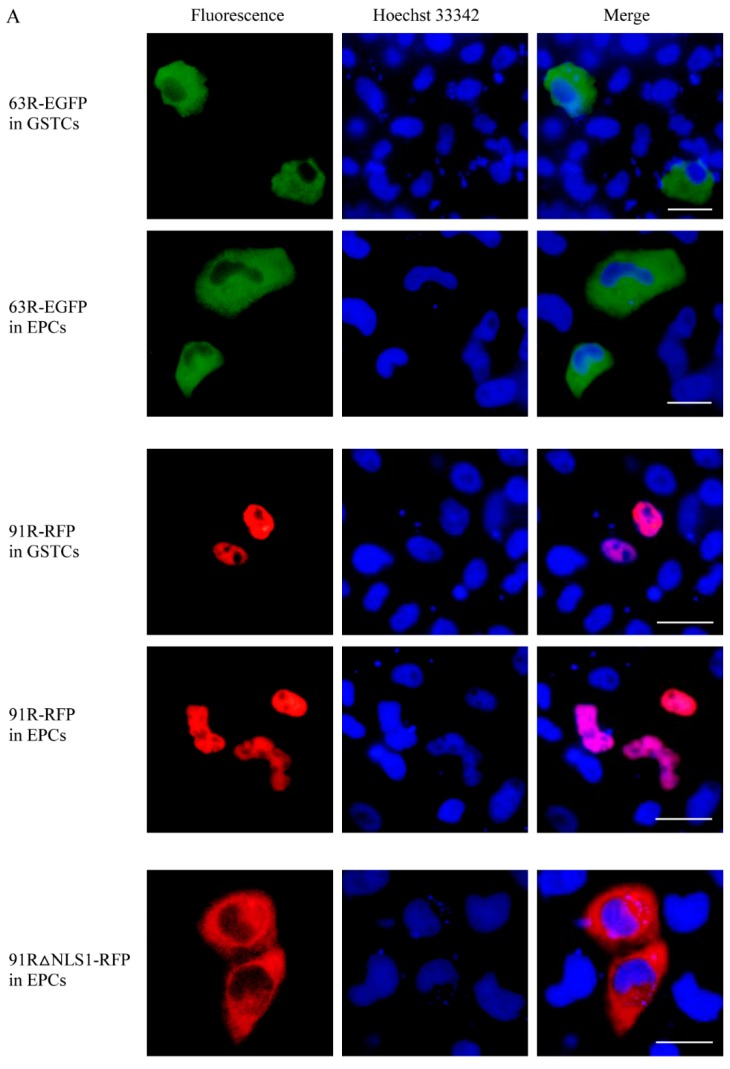
Fluorescence micrographs of cells expressing a single protein or co-expressing two proteins. (**A**) Expressing proteins 63R-EGFP, 91R-RFP, and 91R△NLS1-RFP alone in giant salamander thymus cells (GSTCs) or *Epithelioma papulosum cyprinid* cells (EPCs), respectively. (**B**) Co-expressing two proteins, 63R-EGFP + 91R-RFP and 63R-EGFP + 91R△NLS1-RFP, in GSTCs or EPCs, respectively. 63R-EGFP (green), 91R-RFP (red), 91R△NLS1-RFP (red), nucleus (blue), and colocalization (yellow). Scale bar: 10 μm.

**Figure 4 viruses-11-00416-f004:**
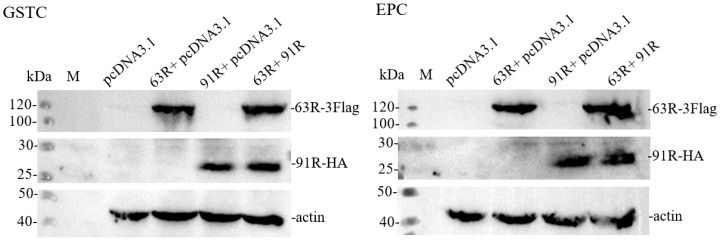
Western blot analysis of protein samples from cells transfected with plasmids for 24 h using anti-Flag and anti-HA. GSTC: in GSTCs expressing protein RGV-63R (63R + pcDNA3.1) or RGV-91R (91R + pcDNA3.1) alone or co-expressing two proteins RGV-63R and RGV-91R (63R + 91R), respectively. EPC: in EPCs expressing protein RGV-63R (63R + pcDNA3.1) or RGV-91R (91R + pcDNA3.1) alone or co-expressing two proteins RGV-63R and RGV-91R (63R + 91R), respectively. The empty vector pcDNA3.1 was used as a control. M: protein molecular mass marker.

**Figure 5 viruses-11-00416-f005:**
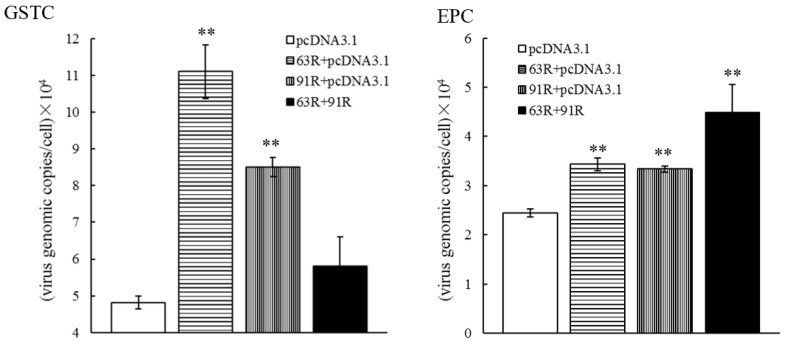
qPCR analysis of RGV genomic copies in GSTCs or EPCs. GSTC: in GSTCs expressing protein RGV-63R (63R + pcDNA3.1) or RGV-91R (91R + pcDNA3.1) alone or co-expressing two proteins RGV-63R and RGV-91R (63R + 91R), respectively. EPC: in EPCs expressing protein RGV-63R (63R + pcDNA3.1) or RGV-91R (91R + pcDNA3.1) alone or co-expressing two proteins RGV-63R and RGV-91R (63R + 91R), respectively. The empty vector pcDNA3.1 was used as a control. The transfected cells were infected with RGV, and RGV genome DNA was extracted from the cells at 48 hpi and quantified by qPCR. Each data point represents the average value of three independent infections. Error bars indicate standard deviations. ** represents *p* < 0.01.

**Table 1 viruses-11-00416-t001:** Primers used in this study.

Name	Sequence (5’ to 3’) ^a^	Usage
*63R*-F	GGGAATTCCATATGGATCTCTTTGTGTACCAGTG (*NdeI*)	pGBKT7-*63R*/Y2H
*63R*-R	CCGGAATTCTTACTTTTTCTTGAACGACA (*EcoRI*)	
*1R*-F	CCGGAATTCATGGCATTCTCGGCAGAAGA (*EcoRI*)	pGADT7-*1R/*Y2H
*1R*-R	CCGCTCGAGTCATAGGGGGGTAAACTTCC (*XhoI*)	
*2L*-F	CCGGAATTCATGTCCATCATCGGAGCGAC (*EcoRI*)	pGADT7-*2L/*Y2H
*2L*-R	CGCGGATCCTTACCATCTCACTGTAGAGA (*BamHI*)	
*9R*-F	GGGAATTCCATATGGAAATGTTTGCATCTAAATC (*NdeI*)	pGADT7-*9R*/Y2H
*9R*-R	CCGGAATTCTCATCGCCACTCAAAGGATT (*EcoRI*)	
*10L*-F	CCGGAATTCATGGACACATCACCCTACGA (*EcoRI*)	pGADT7-*10L/*Y2H
*10L*-R	CCGCTCGAGTCAGGCAAACTTGCCCCTCC (*XhoI*)	
*13L*-F	GGGAATTCCATATGTGCTCCAAACTCGTAGAGAT (*NdeI*)	pGADT7-*13L/*Y2H
*13L*-R	CCGGAATTCTTAGAAACCCATGGTCTCGA (*EcoRI*)	
*16R*-F	CCGGAATTCATGGAACAAGTACCCATAAA (*EcoRI*)	pGADT7-*16R/*Y2H
*16R*-R	CCGCTCGAGCTAATCGTCCAAGTCCGACT (*XhoI*)	
*21R*-F	GGGAATTCCATATGGCTACAAATTACTGTGACGA (*NdeI*)	pGADT7-*21R/*Y2H
*21R*-R	CCGGAATTCTTACATCGTGAAGCTCTCAA (*EcoRI*)	
*23L*-F	CCGGAATTCATGGAAACCATAGTGCTGGT (*EcoRI*)	pGADT7-*23L/*Y2H
*23L*-R	CCGCTCGAGTTACGACGAGGACCCAAATG (*XhoI*)	
*24R*-F	CCGGAATTCATGGCTAACGCTACCATAAA (*EcoRI*)	pGADT7-*24R*/Y2H
*24R*-R	CGCGGATCCCTACTCTTGCTGCTCGGCTC (*BamHI*)	
*29R*-F	GGGAATTCCATATGGCCAATTTTCTACAAGATGT (*NdeI*)	pGADT7-*29R/*Y2H
*29R*-R	CCGGAATTCTCAATGACGCTCCTTGGCCC (*EcoRI*)	
*40R*-F	CCGGAATTCATGCAAGTTTTTCTAGATTT (*EcoRI*)	pGADT7-*40R/*Y2H
*40R*-R	CGCGGATCCTCACCTCCTCCTGCTCCTGC (*BamHI*)	
*44R*-F	GGGAATTCCATATGAGAGTCGTGGTAAACGCAAA (*NdeI*)	pGADT7-*44R/*Y2H
*44R*-R	CCGGAATTCTCACATCAGAAGAGACACGT (*EcoRI*)	
*53R*-F	CCGGAATTCATGGGAGCAGCGGAATCTAT (*EcoRI*)	pGADT7-*53R/*Y2H
*53R*-R	CCGCTCGAGTTAACCCCTGTGGGCCGGAA (*XhoI*)	
*60R*-F	GGGAATTCCATATGGCAATGGTTTCCAACGTAAA (*NdeI*)	pGADT7-*60R/*Y2H
*60R*-R	CCGGAATTCCTACAGGCTCTTTAGGATAA (*EcoRI*)	
*63R*-F	GGGAATTCCATATGGATCTCTTTGTGTACCAGTG (*NdeI*)	pGADT7-*63R/*Y2H
*63R-R2*	CCGGAATTCTTACTTTTTCTTGAACGACA (*EcoRI*)	
*65L*-F	CGCGGATCCTGTCCAGGGGCATGACTACC (*BamHI*)	pGADT7-*65L*/Y2H
*65L*-R	CCGCTCGAGTCACTTGAAGGCTATGGAAA (*XhoI*)	
*73L*-F	CCGGAATTCATGTTTCCTCACGTCACCAT (*EcoRI*)	pGADT7-*73L*/Y2H
*73L*-R	CGCGGATCCTTAGATGTCCAGGGGTTCGT (*BamHI*)	
*87L*-F	CCGGAATTCATGGAAGGTTGGTTGGGAAA (*EcoRI*)	pGADT7-*87L/*Y2H
*87L*-R	CCGCTCGAGCTAGACTCCCTTGGCATGAA (*XhoI*)	
*88R*-F	CCGGAATTCATGTCTTTTCAGAGAGATTA (*EcoRI*)	pGADT7-*88R*/Y2H
*88R*-R	CCGCTCGAGCTACCTGGTCCACCTCTTGC (*XhoI*)	
*91R*-F	GGGAATTCCATATGCTGTGGGAAGCCGTAACAGA (*NdeI*)	pGADT7-*91R*/Y2H
*91R*-R	CCGGAATTCTTAGCCCTCAAAGAGAGTCA (*EcoRI*)	
*92R*-F	GGGAATTCCATATGAGCATCCCTACAGTCATAGC (*NdeI*)	pGADT7-*92R/*Y2H
*92R*-R	CCGGAATTCTTACCGCACATTTCTAGACA (*EcoRI*)	
*95R*-F	CCGGAATTCATGCACGGTTGCAATTGTAA (*EcoRI*)	pGADT7-*95R/*Y2H
*95R*-R	CCGCTCGAGTCAGTTAAAAGTGCTCGTAT (*XhoI*)	
*97R*-F	CCGGAATTCATGTCTTCTGTAACTGGTTC (*EcoRI*)	pGADT7-*97R/*Y2H
*97R*-R	CCGCTCGAGGACCCATGACGGAAAAGACT (*XhoI*)	
*98R*-F	CCGGAATTCATGGCAAACTTTGTGACAGA (*EcoRI*)	pGADT7-*98R/*Y2H
*98R*-R	CCGCTCGAGTTAGGCTCTGACCACAAACA (*XhoI*)	
*101L*-F	CCGGAATTCATGGATCCAGAAGGAATGAT (*EcoRI*)	pGADT7-*101L*/Y2H
*101L*-R	CCGCTCGAGTCACAGCACCTTTCTCAGGT (*XhoI*)	
*102R*-F	CCGGAATTCATGGGCATAAAAGGACTGAA (*EcoRI*)	pGADT7-*102R*/Y2H
*102R*-R	CCGCTCGAGTCACTTGCGCTTGCACTTCT (*XhoI*)	
*63R*-F2	CCCAAGCTTATGGATCTCTTTGTGTACCA (*HindIII*)	pcDNA3.1-*63R-3Flag*
*63R-3Flag*-R	CCGGAATTCTTACTTATCGTCGTCATCCTTGTAATCGATCTTATCGTCGTCATCCTTGTAATCTCCCTTATCGTCGTCATCCTTGTAATCCTTTTTCTTGAACGACACAA (*EcoRI*)	
*91R*-F2	CCCAAGCTTATGCTGTGGGAAGCCGTAAC (*HindIII*)	pcDNA3.1-*91R-HA*
*91R-HA*-R	CCGGAATTCTTAAGCGTAATCTGGAACATCGTATGGGTACATGCCCTCAAAGAGAGTCACGG (*EcoRI*)	
*63R*-R3	CCGGAATTCGACTTTTTCTTGAACGACAC (*EcoRI*)	pEGFP-*63R/* colocalization
*91R*-F3	CCCAAGCTTCAATGCTGTGGGAAGCCGTA (*HindIII*)	pDsRed2-*91R*/ colocalization
*91R*-R2	ACGCGTCGACTTAGCCCTCAAAGAGAGTCA (*SalI*)	
*91R**△**NLS1*-R	CCCATGAGCCTCAGCGTCACGTAGCTGGTAAAGACCGATG	pDsRed2-*91R△NLS1*
*91R**△**NLS1*-F	CATCGGTCTTTACCAGCTACGTGACGCTGAGGCTCATGGG	
*91R**△**NLS2*-R	GAAAAGGGTCCCATCTTGACCACTCCTCCGGACGCCACCA	pDsRed2-*91R△NLS2*
*91R**△**NLS2*-F	TGGTGGCGTCCGGAGGAGTGGTCAAGATGGGACCCTTTTC	
*MCP*-F	ATGGTTGTGGAGCAGGTG	qPCR
*MCP*-R	TGACGCAGGTGTAATTGGAG	

^a^ Sequences of restriction sites are underlined.

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
