# Peer review of "Interaction between Two Iridovirus Core Proteins and Their Effects on Ranavirus (RGV) Replication in Cells from Different Species"

_viruses, 2019, doi:10.3390/v11050416_

Round 1

Reviewer 1 Report

This manuscript describes the characterization of the RGV polymerase and DNA clamp subunits, examines their potential interactions through yeast two-hybrid, co-localization and co-IP assays and assays the roles of these proteins in viral DNA replication.

This manuscript suffers form extensive grammatical errors. Please refrain from using contractions in your text and have it proof-read/edited by an English speaker. At its present state, this manuscript is not of publication quality. 

I feel that that aside from the poor grammar and phrasing, there are significant problems with this work, as described below.

Abstract:

Lines:

22: change ‘distributed’ to ‘localized’.

22-23: rephrase sentence “However, when co-localization…”

24-25: rephrase sentence “RGV-91R deleting…”

Introduction:

Lines:

33: Reword “huge”

34: infectious and pathogen is synonymous 

35: it is inherent infectious pathogen viruses cause viral disease. This does not need to be stated. Please have an English speaker edit the manuscript such statements such as this one are removed or edited. 

47: DNA polymerases do not assemble nucleotides; they catalyze DNA synthesis. Intuitively, viral polymerases are involved in DNA replication. 

Much of the introduction reads like an undergraduate student wrote it. The content therein is not of publication caliber and does not adequately summarize the publications preceding this work.

Materials and Methods:

The authors are inconsistent in the fonts used. 

Change ‘fluorescence observations’ to ‘microscopy’.

Results:

Fig. 1:

Why are there fewer colonies on the enriched plates for 91R, 97R, 98R, 102R than the controls?

The text denoting the yeast crosses is not aligned.

The yeast two-hybrid system is notorious for yielding false positives while the 91R did not generate many colonies. I would like to see repeated experiments with means +/- SEM of colonies for this construct.

Fig. 2:

The blot is missing a marker. 

Fig. 3:

A is cut off.

Get rid of phrasing “ Fluorescent observations”. 

It is not clear what the panels left to right are. These need to be labeled. What are the middle panels?

What is their transfection efficiency? This needs to be reported.  

It is not enough to show a single cell and suggest that this is representative data. The authors need to either provide multiple images or numerical data, reflecting the observations. This is especially true, since the EGFP signal in GSTC cells looks like it may be nuclear as well as cytosolic. 

Fig. 4:

It is not clear why a DNA clamp would increase the amount of viral DNA. It is not clear why a combination of a viral DNA Pol and interacting clamp, which are normally involved in DNA replication, would result in less DNA copies in GSTC cells. The explanation for this observation is not sound, as it is not intuitive that viral ‘preservation’ in the GSTC cells in a function of DNA replication. 

The authors need to provide empirical evidence that the respective constructs are being expressed and to what levels or what proportion of their cells are expressing these proteins. The results in Fig. 3A show at the most a 10% transfection efficiency so the probability that the co-transfections are successful are not very high. The authors need to report A) quantitative expression of 63R, 91R or 63R + 91R and B) demonstrate that a significant proportion of their cells are co-expressing both proteins in the same cells. 

Author Response

Dear editor:

Thank you for the useful and constructive comments on the manuscript (No. viruses-465449). The MS had been rechecked, carefully revised and some new data were added. Such as three parallel experiments of Y2H were performed (Figure 1); complete figure contain the western blot was provided in Figure 2; Fluorescence images that contains two or more positive cells were provided in Figure 3 to make the results more representative; the protein expressions of the constructs were confirmed by western blot analysis in Figure 4. The main revised parts were listed below and marked in red font in viruses-465449R1.

The grammatical mistakes in MS were carefully checked and corrected. The contractions in the text was changed to full name.

Abstract:

Lines of viruses-465449R1:

20: The “distributed” was change to “localized”.

20-21: The sentence “However, when co-localization…” was replaced with “However, when co-expressed, RGV-63R localized in both cytoplasm and …”

22-23: The sentence “RGV-91R deleting…” was replaced with “91RNLS which represents RGV-91R deleting nuclear localization signal localized in cytoplasm and …”

Introduction:

Lines of viruses-465449R1:

31: The word “Huge” was changed to “A large”.

32: The word “infectious” was deleted. 

33: The words “cause severe viral diseases of” was replaced by “infect”.

45: The words “assembling nucleotides” was replaced by “catalyzing DNA synthesis”.

Materials and Methods:

Consistent font-size were used.

All the “Fluorescent observations” in viruses-465449R1 were replaced by “Fluorescence microscopy”

Results:

Figure 1 in viruses-465449R1:

Three parallel experiments for each sample were performed. Yes, false positives might occur in Y2H screens, so we performed co-immunoprecipitation (co-IP) and colocalization assays to confirm protein-protein (RGV-91R and RGV-63R) interaction.

Figure 2 in viruses-465449R1:

Complete figure contain marker was provided.

Fig.3 in viruses-465449R1:

The figure was completely shown. 

The panels were labeled on the top of the figure.

It is estimated 10% of the transfection efficiency by counting of EPC or GSTC cells transfected with pEGFP-63R or pDsRed2-91R.

The images were replaced by that contain two or more cells to make them more representative and more apparent.

Fig.4 in viruses-465449R1:

Information and references about the DNA clamp and DNA pol etc. have been presented in the section of discussion.
  To provide evidence that the respective constructs are being expressed, the expressions of 63R, 91R or 63R + 91R were confirmed by western blot analysis. The results showed that 63R and 91R are both successfully expressed in GSTC or EPC cells, as shown in Figure 4 of MS-R1.

We are looking forward to hearing from you as soon as possible. Many thanks again!

Best wishes,

Qi-Ya Zhang Ph.D. Prof.

Reviewer 2 Report

Review for viruses-465449

Summary: The authors seek to understand whether the products of two core genes (91R and 63R) of the Ranavirus genome interact to aid viral DNA replication. Several molecular virology methods are used to address this issue. First, a Y2H assay shows that the two genes indeed bind, although only white colonies were observed. To follow up, a co-IP assay with a Western blot were performed, which confirmed the binding affinity of the two proteins. To understand co-localization and transport into the nucleus, the authors used a fluorescence experiment. This showed that, when expressed alone, 91R localizes in the nucleus and 63R localizes in the cytoplasm; however, co-expression causes co-localization in the nucleus. The authors further show that transport into the nucleus is likely modulated by NLSs of 91R. Finally, and most interestingly, using an experiment in two cell lines, the authors showed that co-expression of these genes only enhanced viral replication in EPC cells, but hindered replication in GSTC cells. The authors hypothesize that this could be an effect of adaptation of the virus to EPC cell lines over time in the lab.

Impressions: I found this study to be straightforward, but still quite interesting, especially the results of the viral replication experiment in the two cell lines. I have limited comments and believe that this manuscript would fit nicely into Viruses and this special issue with minor changes.

Comments: 

My main comment is that I was disappointed by the brief discussion about the results related to the viral replication experiment in the two cell lines. To me, this is the most interesting finding, and it is quite perplexing. The authors propose that adaptation to the EPC cell lines could be the reason for this effect, but that does not explain the seeming maladaptation to GSTC cell lines. I would like the authors to more fully explore this finding. Specifically, if the two proteins still co-localize in the nuclei of GSTC cells, why would this co-localization inhibit viral replication? Is there a host cell response within the nucleus that might differ between the cell lines? Could the authors draw upon literature in other systems to explain this effect?

Another comment is that the hypotheses of this study were not well explained in the introduction, and there was a disconnect between hypotheses and the methods. In other words, it wasn’t immediately clear why the authors were using the specific methods that they list in lines 64-66. In addition, to reach a broader audience, I think that the authors should very briefly introduce each of their assays (e.g., Y2H, co-IP, etc.) in their introduction and state what these methods are used to test. This may be resolved by better linking the hypotheses to the methods (e.g., We used Y2H to test the hypothesis that the gene products can physically bind to each other, while a fluorescence experiment was used to test whether the gene products had similar co-localization in different host cells, etc.)

Finally, and will all due respect to the authors and their fine quality of scientific work, the manuscript needs substantial editing for English style and grammar. I could still understand the authors’ main points, but there are many places where the message could be clarified.

Minor comment: Could the resolution of Figure 1 be enhanced? 

Author Response

Dear editor:

Thank you for the useful and detailed comments and recommendations on the manuscript (No. viruses-465449). The MS had been carefully revised. The revised parts were listed below and marked in red font in viruses-465449R1.

Thanks very much for your useful suggestions on the viral replication experiment. The possible explanations for the phenomenon were added in the discussion section (lines 286-293 of viruses-465449R1). The interesting phenomenon is an issue that we try to design more experiments to confirm and explain the mechanism.

The description about the methods (Y2H, fluorescence, co-IP) has been added in the introduction of viruses-465449R1.

The grammatical mistakes were carefully checked and corrected.

Three parallel experiments for each sample were performed and the resolution of Figure 1 was enhanced.

We are looking forward to hearing from you as soon as possible. Many thanks again!

Best wishes,

Qi-Ya Zhang Ph.D. Prof.

Round 2

Reviewer 1 Report

Contrary to the authors’ comments that “the grammatical mistakes in MS were carefully checked and corrected” there are still many grammatical issues with this work. For example, the first sentence of the introduction reads “A large amount of aquatic viruses regulate population dynamics and community interactions in aquatic ecosystem”. There are several grammatical errors in this sentence alone.

Unfortunately, this manuscript does not present the use of written English, which would be suitable for publication.

The manuscript now has two Figures labeled as Figure 4. 

The authors failed to indicate and address each of my previous suggestions and concerns. Chiefly amongst these was the fact that this manuscript fails to provide numerical representation of the results presented in Fig. 3. As the authors themselves admit, there is a very low transfection efficiency of their individual constructs, which begs the question of what proportion of the cells were doubly transfected. This was not addressed and the comments ignored. I find it difficult to believe (as would be indicated by Fig. 4) that the majority of 91R co-precipitated with 63R, considering the low transfection efficiency of individual constructs and the lack of information about what proportions of the constructs were successfully co-transfected. 

Please have your manuscript professionally edited and upon resubmission, please include my original comments and suggestions, with individual point-by-point address of each of the comments.

Why was one of the authors removed from the manuscript?

Author Response

We really appreciate your timely review of our manuscript (viruses-465449). Your comments and suggestions have been instrumental in the revision of the manuscript. The manuscript has been revised extensively according to your comments. The main revised parts were listed below and marked in red font in viruses-465449R2.

Comments: Contrary to the authors’ comments that “the grammatical mistakes in MS were carefully checked and corrected” there are still many grammatical issues with this work. For example, the first sentence of the introduction reads “A large amount of aquatic viruses regulate population dynamics and community interactions in aquatic ecosystem”. There are several grammatical errors in this sentence alone.

Response: The grammatical issues in the viruses-465449R2 have been rechecked and corrected with the help of the English editing service of the MDPI. 

CommentsUnfortunately, this manuscript does not present the use of written English, which would be suitable for publication.

Response: The MS of viruses-465449R2 was edited by the English editing service of the MDPI.

Comments: The manuscript now has two Figures labeled as Figure 4. 

Response: The Figure 4 showing the results of qPCR was labeled as Figure 5 in viruses-465449R2.

Comments: The authors failed to indicate and address each of my previous suggestions and concerns. Chiefly amongst these was the fact that this manuscript fails to provide numerical representation of the results presented in Fig. 3. As the authors themselves admit, there is a very low transfection efficiency of their individual constructs, which begs the question of what proportion of the cells were doubly transfected. This was not addressed and the comments ignored. I find it difficult to believe (as would be indicated by Fig. 4) that the majority of 91R co-precipitated with 63R, considering the low transfection efficiency of individual constructs and the lack of information about what proportions of the constructs were successfully co-transfected. 

Response: The transfection efficiency was about 10% by counting the number of EPC or GSTC cells transfected with plasmids pEGFP-63R. The plasmids pcDNA3.1-63R-3Flag (0.25 μg) and pcDNA3.1-91R-HA (0.25 μg) were mixed when they were cotransfected into cells. There is about 7.6% cells cotransfected with plasmids pcDNA3.1-63R-3Flag and pcDNA3.1-91R-HA, and about 2.4% cells transfected with alone plasmid by counting the number of transfected cells using immunofluorescence. The transfection efficiency was added in lines 215-219 of viruses-465449R2.

Comments: Please have your manuscript professionally edited and upon resubmission, please include my original comments and suggestions, with individual point-by-point address of each of the comments.

Response: The manuscript (viruses-465449R2) has been professionally edited with the help of the English editing of the journal Viruses. Your original comments and suggestions were included in response R2.

Comments: Why was one of the authors removed from the manuscript?

Response: Chinchar VG, one of authors in last manuscript (viruses-465449), considered that his name should not been added in the author list, since he is an editor of the journal Viruses. So, the name was removed from the manuscript.

We are looking forward to hearing from you as soon as possible. Thank you very much.

Best wishes,

Qi-Ya Zhang Ph.D. Prof.